# Sample as you Infer:
# Predictive Coding with Langevin Dynamics

## Abstract

We present a novel algorithm for parameter learning in generic deep generative models that builds upon the predictive coding (PC) framework of computational neuroscience. Our approach modifies the standard PC algorithm to bring performance on-par and exceeding that obtained from standard variational auto-encoder (VAE) training. By injecting Gaussian noise into the PC inference procedure we re-envision it as an overdamped Langevin sampling, which facilitates optimisation with respect to a tight evidence lower bound (ELBO). We improve the resultant encoder-free training method by incorporating an encoder network to provide an amortised warm-start to our Langevin sampling and test three different objectives for doing so. Finally, to increase robustness to the sampling step size and reduce sensitivity to curvature, we validate a lightweight and easily computable form of preconditioning, inspired by Riemann Manifold Langevin and adaptive optimizers from the SGD literature. We compare against VAEs by training like-for-like generative models using our technique against those trained with standard reparameterisation-trick-based ELBOs. We observe our method out-performs or matches performance across a number of metrics, including sample quality, while converging in a fraction of the number of SGD training iterations.

## 1 Introduction

In recent decades the Bayesian brain hypothesis has emerged as a compelling general framework for understanding perception and learning in the brain (Pouget et al., 2013; Clark, 2013; Kanai et al., 2015). Under this framework, the brain is posited as encoding a probabilistic generative model engaged in a joint scheme of inference over the hidden causes of its observations and learning over its model parameters. One of the most popular instantiations of this view is predictive coding (PC), a computational scheme which employs hierarchical latent Gaussian generative models with complex, non-linear conditional parameterizations. In recent years, PC has garnered substantial attention for its potential to elucidate cortical function (Rao & Ballard, 1999; Friston, 2018; Mumford, 1992; Hosoya et al., 2005; Hohwy et al., 2008; Bastos et al., 2012; Shipp, 2016; Feldman & Friston, 2010; Fountas et al., 2022). Despite its predictive appeal in the cognitive sciences, the practical applicability and performance of PC in training deep generative models, akin to those conjectured to operate in the brain, has yet to be fully realized (Zahid et al., 2023a).

Concurrent to these developments in the cognitive sciences, a separate revolution has been occurring in the statistical literature driven by the use of gradient-based Monte Carlo sampling methods such as Hamiltonian Monte Carlo (HMC) (Roberts & Tweedie, 1996; Neal, 2011; Hoffman & Gelman, 2011; Girolami & Calderhead, 2011; Ma et al., 2019). These methods facilitate the sampling of intractable distributions through the intelligent construction of Markov chains with proposals informed by gradient information from the log density being sampled. Notably, one of the simplest algorithms within this class is the overdamped Langevin algorithm (Rossky et al., 1978; Roberts & Tweedie, 1996; Roberts & Rosenthal, 1998), which admits an interpretation as both a limiting case of HMC, and as a discretisation of a Langevin diffusion (Neal, 2011).

This paper introduces several advancements aimed at extending the predictive coding framework using techniques from gradient-based Markov Chain Monte Carlo (MCMC) for use in training deep generative models:

- We show that by injecting appropriately scaled Gaussian noise, the standard PC inference procedure may be interpreted as an (unadjusted) overdamped Langevin sampling.
- Utilizing these Langevin samples, we compute gradients with respect to a tight evidence lower bound (ELBO), which we then optimize our model parameters against.
- To improve chain mixing time, we train approximate inference networks for amortized warm-starts and evaluate three distinct objectives for their optimization.
- We investigate and validate a light-weight diagonal preconditioning strategy for increasing robustness to the Langevin step size, inspired by adaptive optimization techniques.

## 2 METHODOLOGY

### 2.1 INFERENCE AS LANGEVIN DYNAMICS

The standard PC recipe for inference and learning under a generative model, for static observations, may be described succinctly as follows (Rao & Ballard, 1999; Bogacz, 2017; Millidge et al., 2020):

1. Define a (possibly hierarchical) graphical model over latent ($\mathbf{z} \in \mathbb{R}^d$) and observed ($\mathbf{x} \in \mathbb{R}^n$) states with parameters $\boldsymbol{\theta}$: $\log p(\mathbf{x}, \mathbf{z}|\boldsymbol{\theta})$

2. For each observation $\boldsymbol{x}^{(i)} \sim \mathcal{D}$, where $\mathcal{D}$ is the data-generating distribution.

   **Inference:** Iteratively enact a gradient ascent on $\log p(\boldsymbol{x}^{(i)}, \boldsymbol{z}|\theta)$ with respect to latent states ($\boldsymbol{z}$)

   $$\boldsymbol{z}^{(t)} = \boldsymbol{z}^{(t-1)} + \gamma \nabla_{\boldsymbol{z}} \log p(\mathbf{x}^{(i)}, \mathbf{z}^{(t-1)}|\boldsymbol{\theta}) \tag{1}$$

   Until you obtain an MAP estimate: $\boldsymbol{z}_{\text{MAP}} = \max_{\boldsymbol{z}} \log p(\mathbf{x}^{(i)}, \mathbf{z}|\boldsymbol{\theta})$

   **Learning:** Update model parameters $\theta$ using stochastic gradient descent with respect to the log joint evaluated at the MAP (averaged over multiple observations if using mini-batches):

   $$\boldsymbol{\theta}^{(i)} = \boldsymbol{\theta}^{(i-1)} + \alpha \nabla_{\boldsymbol{\theta}} \log p(\boldsymbol{x}^{(i)}, \boldsymbol{z}_{\text{MAP}}|\boldsymbol{\theta}^{(i-1)}) \tag{2}$$

One simple and relevant framing of this process is that of a variational ELBO maximising scheme under the assumption of a Dirac delta (point-mass) approximate posterior (Friston, 2003; 2005; Friston & Kiebel, 2009; Zahid et al., 2023b). In practice, the restrictiveness of this Dirac delta posterior significantly impairs the quality of the resultant model due to the expected divergence between the true model posterior and the Dirac delta function situated at the MAP estimate. Indeed, previous attempts at reducing the severity of this assumption, by adopting quadratic approximations to the posterior at the MAP, (Zahid et al., 2023b), succeeded in improving model quality to a degree, but suffered from high computational cost while still performing significantly worse than their variational auto-encoder counterparts.

Our contribution begins with the observation that by injecting appropriately scaled Gaussian noise into Equation 1, one obtains an unadjusted Langevin algorithm (ULA). Specifically, the ULA may be considered the discretisation of a continuous-time Langevin diffusion (Rossky et al., 1978; Roberts & Tweedie, 1996), characterised by the following stochastic differential equation,

$$d\boldsymbol{Z}_t = -\nabla_{\boldsymbol{z}} U(\boldsymbol{Z}_t)dt + \sqrt{2}d\boldsymbol{W}_t \tag{3}$$

where $\boldsymbol{W}_t$ is a d-dimensional Brownian motion and admits a unique invariant density equal to $\frac{e^{-U(\boldsymbol{z})}}{\int_{\mathbb{R}^d} e^{-U(\boldsymbol{z})}dz}$ under mild conditions. Setting the potential energy ($U(\boldsymbol{z})$) to $-\log p(\mathbf{x}^{(i)}, \mathbf{z}|\boldsymbol{\theta})$, for an observation $\mathbf{x}^{(i)}$ gives us:

$$d\boldsymbol{Z}_t = \nabla_{\boldsymbol{z}} \log p(\mathbf{x}^{(i)}, \boldsymbol{Z}_t|\boldsymbol{\theta})dt + \sqrt{2}d\boldsymbol{W}_t \tag{4}$$

for which the corresponding Euler–Maruyama discretisation scheme is:

$$\boldsymbol{z}^{(t)} = \boldsymbol{z}^{(t-1)} + \gamma \nabla_{\boldsymbol{z}} \log p(\boldsymbol{x}^{(i)}, \boldsymbol{z}^{(t-1)}|\boldsymbol{\theta}) + \sqrt{2\gamma}\eta \tag{5}$$

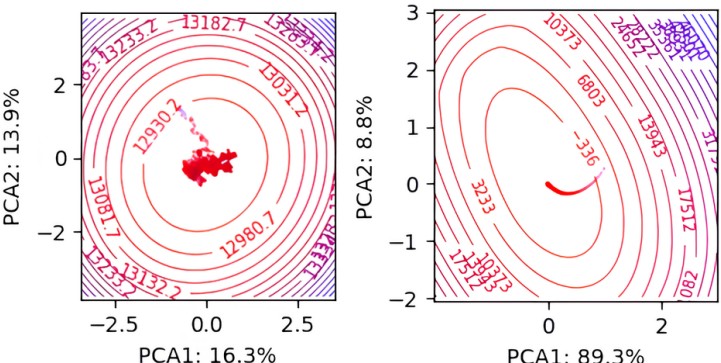

Figure 1: Projection of high-dimensional latent state trajectories under standard PC inference (right), and Langevin PC sampling (left), using normalised PCA trajectories. Latent state dynamics under Langevin PC result in a principled exploration of the posterior. More examples trajectories, and further details on how these were computed may be found in Appendix A.2. Contour lines and hue correspond to values of the negative log joint probability (blue high, red low), marker brightness corresponds to time-step (earlier is lighter).

with $\eta \sim \mathcal{N}(\mathbf{0}, \boldsymbol{I})$. This is simply equal to a standard PC inference iteration (Equation 1) with the addition of some scaled Gaussian noise. With the inclusion of this Gaussian noise, the resultant iterates $\boldsymbol{z}^{(t)}$ would thus be interpretable as samples of the true model posterior, as $t \to \infty$, up to a bias induced by discretisation (Besage, J. E, 1994; Roberts & Tweedie, 1996).

Next, we note that by treating the (biased) samples from our Langevin chain as samples from an approximate posterior instead, we may compute gradients of a Monte Carlo estimate for the evidence lower-bound with respect to our model parameters $\boldsymbol{\theta}$:

$$\nabla_{\boldsymbol{\theta}} \mathcal{L}_{\text{ELBO}}(\mathbf{x}) = \nabla_{\boldsymbol{\theta}} \left[ \mathbb{E}_{\tilde{p}(\mathbf{z}|\mathbf{x})}[\log p(\mathbf{x}, \mathbf{z}|\boldsymbol{\theta})] + \mathbb{E}_{\tilde{p}(\mathbf{z}|\mathbf{x})}[\log \tilde{p}(\mathbf{z}|\mathbf{x})] \right] \tag{6}$$

Where the approximate posterior $\tilde{p}(\mathbf{z}|\mathbf{x})$ corresponds to the empirical distribution of our Langevin chain. Because we are only interested in gradients with respect to our parameters $\boldsymbol{\theta}$, the intractable entropy term of our sample distribution may be ignored

$$= \nabla_{\boldsymbol{\theta}} \left[ \mathbb{E}_{\tilde{p}(\mathbf{z}|\mathbf{x})}[\log p(\mathbf{x}, \mathbf{z}|\boldsymbol{\theta})] \right] + \underbrace{\nabla_{\boldsymbol{\theta}} \left[ \mathbb{E}_{\tilde{p}(\mathbf{z}|\mathbf{x})}[\log \tilde{p}(\mathbf{z}|\mathbf{x})] \right]}_{=0} \tag{7}$$

$$\approx \nabla_{\boldsymbol{\theta}} \frac{1}{T} \sum_t \log p(\boldsymbol{x}, \boldsymbol{z}^{(t)}|\boldsymbol{\theta}) \tag{8}$$

Crucially, optimisation of this ELBO simply requires computing the gradient of our negative potential energy $\log p(\boldsymbol{x}, \boldsymbol{z}|\boldsymbol{\theta})$, with respect to $\boldsymbol{\theta}$ rather than $\boldsymbol{z}$, and is (computationally) identical to the learning step in Equation 2. From the perspective of neurobiological plausibility, this result is a pleasant surprise, as there already exists a substantial literature on how the dynamics described by Equation 1 and 2 may be implemented neuronally (Friston, 2003; 2005; Shipp, 2016; Bastos et al., 2012). Thus, the Langevin PC algorithm demands no additional neurobiological machinery other than the injection of Gaussian noise into our standard PC iterates. We briefly discuss the possible implications of this in Section 4.

From the perspective of an in-silico implementation, these gradients may be collected iteratively as the Markov chain is constructed, resulting in constant memory requirements independent of the chain length $T$, while reusing portions of the same backward pass used to compute our Langevin drift: $\nabla_{\boldsymbol{z}} \log p(\boldsymbol{x}, \boldsymbol{z}|\boldsymbol{\theta})$.

## 2.2 AMORTISED WARM-STARTS

It is well-known that MCMC sampling methods, while powerful in theory, are notoriously sensitive to their choice of hyperparameters in practice (Steve Brooks, Andrew Gelman, Galin Jones, Xiao-Li

Meng, 2011). One such choice is the state of initialisation for a Markov chain. A poor initialisation, far from the typical set of the invariant density will result in an inefficient chain with poor mixing time. This is of particular importance if we require constructing this Markov chain within each SGD training iteration. Traditional strategies to ameliorate this issue generally appeal to burn-in, i.e the discarding of a series of initial samples (Andrew Gelman et al., 2015), or by initialising at the MAP found via numerical optimisation (Salvatier et al., 2015). Such strategies are costly, particularly for our Langevin dynamics as they require expensive and wasted network evaluations.

We resolve this issue by training an amortised warm-up model (equivalently, an approximate inference model) conditional on observations. This allows us to provide a warm-start to our Langevin chain that is ideally within the typical set. Architecturally this network may be chosen to resemble standard encoders, in encoder-decoder frameworks such as the VAE (Kingma & Welling, 2014), however the availability of (biased) samples from the model posterior obtained through Langevin dynamics afford us greater flexibility in how we train it. Here we propose and validate three objectives for training our amortised warm-start model: the forward KL, reverse KL and Jeffrey's divergence.

### 2.2.1 FORWARD KL

Given Langevin samples from the model posterior, the most obvious objective for optimising our approximate inference network is the expected forward Kullback–Leibler divergence between the model posterior and our approximate posterior, with expectation approximated with mini-batches of observations. Specifically, the forward KL divergence can be separated into an intractable but encoder-independent entropy term, and a cross entropy term for which we may obtain a Monte Carlo estimate using our Langevin samples:

$$D_{\mathrm{KL}}(\tilde{p}(\mathbf{z}|\mathbf{x})|q(\mathbf{z}|\mathbf{x},\boldsymbol{\phi})) = \mathbb{E}_{(\tilde{p}(\mathbf{z}|\mathbf{x})} \left[ \log \frac{\tilde{p}(\mathbf{z}|\mathbf{x})}{q(\mathbf{z}|\mathbf{x},\boldsymbol{\phi})} \right] \tag{9}$$

where we are exclusively interested in obtaining gradients with respect to $\boldsymbol{\phi}$, and as such:

$$\nabla_{\boldsymbol{\phi}} D_{\mathrm{KL}}(\tilde{p}(\mathbf{z}|\mathbf{x})|q(\mathbf{z}|\mathbf{x},\boldsymbol{\phi})) = \underbrace{\nabla_{\boldsymbol{\phi}} \mathbb{E}_{\tilde{p}(\mathbf{z}|\mathbf{x})} \left[ \log \tilde{p}(\mathbf{z}|\mathbf{x}) \right]}_{0} - \nabla_{\boldsymbol{\phi}} \mathbb{E}_{\tilde{p}(\mathbf{z}|\mathbf{x})} \left[ \log q(\mathbf{z}|\mathbf{x},\boldsymbol{\phi}) \right] \tag{10}$$

$$= \nabla_{\boldsymbol{\phi}} \mathbb{E}_{\tilde{p}(\mathbf{z}|\mathbf{x})} \left[ \log q(\mathbf{z}|\mathbf{x},\boldsymbol{\phi}) \right] \tag{11}$$

which is simply the cross-entropy between our empirical Langevin posterior distribution and our approximate inference model. We will denote the Monte Carlo estimate for this approximate inference objective for a mini-batch of observations and a single batch of their associated posterior samples, as $\mathcal{L}_{A_F}(\boldsymbol{x}, \boldsymbol{z})$.

### 2.2.2 REVERSE KL

While the forward KL is an enticing objective given our access to samples from the posterior, its well-known moment matching behaviour may result in an initialisation at the average of multiple modes and as such a low posterior probability, particularly given the Gaussian approximate posterior we will be adopting (Bishop, 2006). In such circumstances, the mode matching behaviour of the reverse KL may be more appropriate. Computing the reverse KL divergence directly is difficult given our inability to directly evaluate the true log posterior probability. We can circumvent this by appealing to the standard ELBO, evaluated using the reparameterisation trick of Kingma & Welling (2014), which admits a decomposition consisting of an encoder-independent model evidence term, and the reverse KL we wish to obtain gradients from,

$$\mathcal{L}_{A_R} = D_{\mathrm{KL}}(q(\mathbf{z}|\mathbf{x},\boldsymbol{\phi})|p(\mathbf{z}|\mathbf{x})) = \mathbb{E}_{q(\mathbf{z}|\mathbf{x},\boldsymbol{\phi})} \left[ \log \frac{q(\mathbf{z}|\mathbf{x},\boldsymbol{\phi})}{p(\mathbf{z}|\mathbf{x})} \right] \tag{12}$$

where we are once again exclusively interested in obtaining gradients with respect to $\boldsymbol{\phi}$, and as such,

$$\nabla_{\boldsymbol{\phi}} D_{\mathrm{KL}}(q(\mathbf{z}|\mathbf{x},\boldsymbol{\phi})|p(\mathbf{z}|\mathbf{x})) = \nabla_{\boldsymbol{\phi}} \left[ D_{\mathrm{KL}}(q(\mathbf{z}|\mathbf{x},\boldsymbol{\phi})|p(\mathbf{z}|\mathbf{x})) - \log p(\mathbf{x}) \right] \tag{13}$$

$$= \nabla_{\boldsymbol{\phi}} \left[ -\mathbb{E}_{q(\mathbf{z}|\mathbf{x},\boldsymbol{\phi})}[\log p(\mathbf{x}|\mathbf{z})] + D_{\mathrm{KL}}(q(\mathbf{z};\boldsymbol{\phi})|p(\mathbf{z})) \right] \tag{14}$$

$$= \nabla_{\boldsymbol{\phi}} \mathcal{L}_{\mathrm{ELBO}} \tag{15}$$

### 2.2.3 JEFFREY'S DIVERGENCE

Finally, by averaging gradients from the forward and reverse KL divergences we may optimise with respect to (half) the Jeffrey's divergence, also known as the symmetrised KL (Jeffreys, 1946), which can be shown to upper bound 4 times the Jensen-Shannon divergence (Lin, 1991).

$$\nabla_{\boldsymbol{\phi}} \mathcal{L}_{A_J} = \frac{1}{2} \nabla_{\boldsymbol{\phi}} \left[ D_{\text{KL}}(p(\mathbf{z}|\mathbf{x})|q(\mathbf{z}|\mathbf{x},\boldsymbol{\phi})) + D_{\text{KL}}(q(\mathbf{z}|\mathbf{x},\boldsymbol{\phi})|p(\mathbf{z}|\mathbf{x})) \right] \tag{16}$$

### 2.3 ADAPTIVE PRECONDITIONING

There now exists a sizeable literature approaching gradient-based sampling from the perspective of optimisation in the space of probability measures (Jordan et al., 1998; Wibisono, 2018). This framing has led to the development of analogues to well-known methods from the classical optimisation literature, such as Nesterov's acceleration (Ma et al., 2019). Similar analogues to preconditioning have also emerged in the literature, with (Girolami & Calderhead, 2011), demonstrating that an appropriately chosen, possibly position-specific, preconditioning matrix may be used to exploit the natural Riemannian geometry over the induced distributions, improving mixing time and sampling efficiency. A number of works have subsequently capitalized on this technique with a variety of Riemannian metrics, primarily within the context of stochastic gradient Langevin dynamics (SGLD) - a technique that applies Langevin dynamics to noisy mini-batch gradients over deep neural network parameters to obtain posterior samples (Welling & Teh, 2011; Ahn et al., 2012; Patterson & Teh, 2013; Li et al., 2015).

Here we adopt the adaptive second-moment computation of the Adam (Kingma & Ba, 2017) optimizer as our preconditioning matrix, computed with iterates over the log unnormalised probability $\log p(\mathbf{x}, \mathbf{z}_t)$. The resultant algorithm may be considered analogous to the use of the diagonal RMSProp preconditioner for SGLD by Li et al. (2015), with key differences being in the use of a debiasing step, the use of non-stochastic gradients, and the inclusion of the gradient over the log prior in our second-moment calculations. We note that the Itô SDE associated with an overdamped Langevin diffusion with position-dependent metric tensor $G(\boldsymbol{X}_t)$, may be written as (Girolami & Calderhead, 2011; Ma et al., 2015; Roberts & Stramer, 2002; Xifara et al., 2014):

$$d\boldsymbol{Z}_t = G(\boldsymbol{X}_t)\nabla_z \log p(\mathbf{x}, \mathbf{z}|\boldsymbol{\theta})dt + \Gamma(\boldsymbol{Z}_t)dt + \sqrt{2G(\boldsymbol{Z}_t)}d\boldsymbol{W}_t \tag{17}$$

where the term $\Gamma(\boldsymbol{Z}_t)$ accounts for changes in local curvature of the manifold, and is defined as:[1]

$$\Gamma_i(\boldsymbol{Z}_t) = \sum_j \frac{G_{ij}(\boldsymbol{Z}_t)}{\partial Z_j} \tag{18}$$

The resultant discretization given by the Euler-Murayama scheme follows analogously to that in Equation 5. We follow identically to Ahn et al. (2012) and Li et al. (2015) and choose to ignore the $\Gamma_i(\boldsymbol{X}_t)$ term in our final discretized algorithm; selecting our preconditioning decay rate to be close to 1, such that our manifold changes slowly. Our final preconditioned algorithm with amortised warm-starts is described in Algorithm 1.

## 3 RESULTS

For all experiments considered here, we adopt generative and warm-start models that are largely coincident with the encoder, and decoder respectively from the VAE architecture of Higgins et al. (2016), with minor modifications, adopted from more recent VAE models (Child, 2021; Vahdat & Kautz, 2021), such as SiLU activation functions and softplus parameterised variances. Complete details of model architecture and hyperparameters can be found in Appendix A.1

---

[1] We note that this term appears slightly differently to that found in Roberts & Stramer (2002) and Girolami & Calderhead (2011), as the original formulation was shown by (Xifara et al., 2014) to correspond to the density function with respect to a non-Lebesgue measure (after correcting a transcription error). The term as used in this paper is of the form suggested by (Xifara et al., 2014) which has the required invariant density with respect to the Lebesgue measure.

**Algorithm 1** Preconditioned Langevin PC with Amortized Warm-Starts.

This version corresponds to warm-starts trained with Jeffrey's divergence. For the version corresponding to warm-starts with just the reverse KL, remove the forward KL accumulation and the coefficient of $\frac{1}{2}$ from the reverse KL gradients.

---

**Require:** $\mathcal{D}$ (data-generating distribution)
**Require:** $p(\mathbf{x}, \mathbf{z}|\boldsymbol{\theta})$ (generative model with parameters $\boldsymbol{\theta}$)
**Require:** $q(\mathbf{z}|\mathbf{x}, \boldsymbol{\phi})$ (approximate inference model with parameters $\boldsymbol{\phi}$)
**Require:** $\beta$: Preconditioning decay rate
**Require:** $\gamma, \alpha, T$: Langevin step size, parameter learning rate, and number of sampling steps
    **for** $\mathbf{x} \sim \mathcal{D}$ **do**
        $\boldsymbol{g_\theta}, \boldsymbol{g_\phi}, \boldsymbol{m}^{(0)} \leftarrow \mathbf{0}$
        $\boldsymbol{z}^{(0)} \sim q(\mathbf{z}|\boldsymbol{x}, \boldsymbol{\phi})$              $\triangleright$ Warm-start with approximate inference network
        $\boldsymbol{g_\phi} \mathrel{+}= \frac{1}{2}\nabla_{\boldsymbol{\phi}}\mathcal{L}_{A_R}$           $\triangleright$ Compute reverse KL gradients (Equation 15)
        **for** $t \in \{1, 2, \ldots, T\}$ **do**
            $\boldsymbol{g_z} \leftarrow \nabla_{\boldsymbol{z}} \log p(\mathbf{x}, \boldsymbol{z}^{(t-1)}|\boldsymbol{\theta})$                $\triangleright$ Compute drift
            $\boldsymbol{m}^{(t)} \leftarrow \beta \cdot \boldsymbol{m}^{(t-1)} + (1 - \beta) \cdot (\boldsymbol{g_z^T g_z})$      $\triangleright$ Compute uncorrected second-moment
            $\hat{\boldsymbol{m}}^{(t)} \leftarrow \sqrt{\boldsymbol{m}^{(t)}/(1 - \beta^t)}$               $\triangleright$ Bias correction
            $\boldsymbol{z}^{(t)} \leftarrow \gamma \cdot \boldsymbol{g_z} \oslash \hat{\boldsymbol{m}}^{(t)} + \eta$, where $\eta \sim \mathcal{N}(\mathbf{0}, \text{diag}(2\gamma \cdot \hat{\boldsymbol{m}}))$    $\triangleright$ Compute new iterate
            $\boldsymbol{g_\theta} \mathrel{+}= \nabla_{\boldsymbol{\theta}} \log p(\mathbf{x}, \boldsymbol{z}^{(t-1)}|\boldsymbol{\theta})$       $\triangleright$ Accumulate gradients for expected ELBO
            $\boldsymbol{g_\phi} \mathrel{+}= \frac{1}{2T}\nabla_{\boldsymbol{\phi}}\mathcal{L}_{A_F}$         $\triangleright$ Accumulate forward KL gradients (Equation 11)
        **end for**
        $\boldsymbol{\theta} \mathrel{+}= \alpha \cdot \boldsymbol{g_\theta}$
        $\boldsymbol{\phi} \mathrel{+}= \alpha \cdot \boldsymbol{g_\phi}$
    **end for**

---

## 3.1 Approximate Inference Objectives

We begin by investigating the performance of our three approximate inference objectives, the forward KL, reverse KL and Jeffrey's divergence on the quality of our samples when trained with CIFAR-10 (Krizhevsky, 2009), SVHN (Netzer et al., 2011) and CelebA (64x64) (Liu et al., 2015). As a baseline, we also test with no amortized warm-starts, instead using samples from our prior, for which we adopt an isotropic Gaussian with variance 1, to initialise our Langevin chain. For all tests, we also adopt this prior initialisation for the first 50 batches of training to ameliorate the effects of any poor initialisation in our warm-start models.

To quantify sample quality we compute the both the standard Fréchet distance with Inceptionv3 representations (FID) (Heusel et al., 2017), and with DinoV2 representations (denoted by FDD in this paper), which was shown by Stein et al. (2023) to correlate significantly more closely to human evaluations of sample quality. We observe a largely consistent relationship for the forward KL, with the objective exhibiting both poor performance in terms of sample quality and training instabilities resulting from an increasingly poor initialisation as training progresses. We validate this by recording changes in log probability and the L2 normed gradient of the log probability for random samples during their sampling trajectories for the three objectives. We observe significantly qualitatively different behaviours for the forward KL initialisations, observing drift-dominant conditions with dynamics dominated by maxima-seeking behaviour suggesting poor initialisation far from the mode. Example recordings of the change in log probability $\Delta \log p(\boldsymbol{x}, \boldsymbol{z})$ may be found in Figure 2B. Further examples, and the equivalent normed gradient plots can be found in Appendix A.3.

In comparison we observe clear improvements in sample quality and FID when using amortised warm-starts trained with Jeffrey's divergence or the reverse KL over the baseline encoder-only models. Due to the marginally improved performance of the reverse KL in terms of the FDD and its reduced computational cost - from a lack of forward KL gradient accumulation - we adopt this objective for all subsequent experiments. FDD values for the three objectives can be found in Figure 2, with corresponding FID values in Appendix A.3, note that due to exploding gradients for the forward KL objective at later epochs, the FID and FDD values in for the forward KL correspond to performance at 1 epoch.

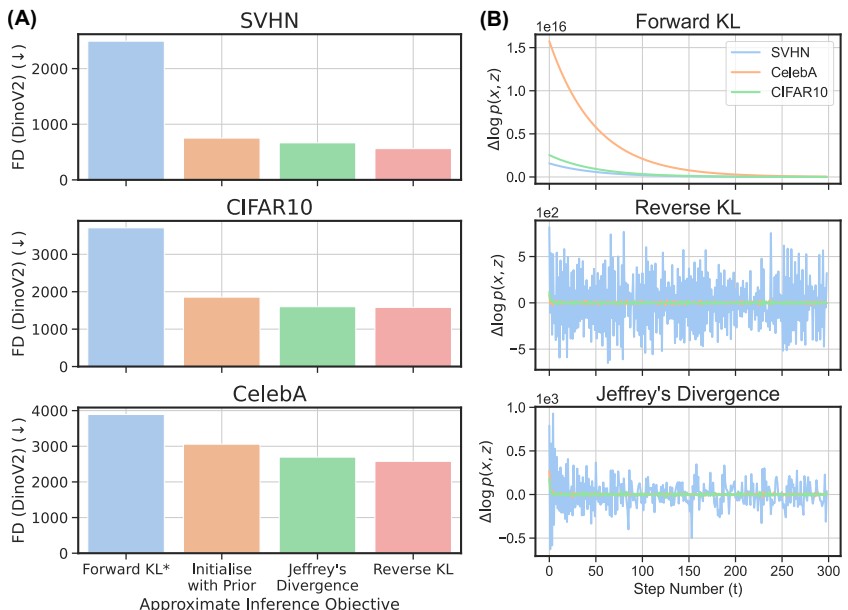

Figure 2: (A) FDD when using amortised warm-starts trained with our three approximate inference objectives, and baseline with no warm-start model, using initialisation with the prior. * Values for the forward KL objective are reported for 1 epoch due to the instability of this objective resulting in exploding gradients. (B) Changes in log probability ($\Delta \log p(\boldsymbol{x}, \boldsymbol{z})$) during Langevin sampling show forward KL initialisation results in long periods of drift-dominant conditions far from the mode.

## 3.2 PRECONDITIONING INDUCED ROBUSTNESS

We assess the impact of preconditioning on increasing step sizes by testing models with and without preconditioning as we vary the Langevin step size from 0.01 to 0.5. We observe a substantial protective effect on the degradation of sample quality as step size increases in terms of both the FDD (Figure 3), and FID (Figure 6 in Appendix A.3) of the resultant models. We also find that while preconditioned models generally exhibit better sample quality over their non-preconditioned counterparts, this trend begins to reverse at the very lowest Langevin step-sizes tested, where non-preconditioned models reach parity or improved performance. For our final set of experiments we evaluate Langevin PC without preconditioning at a fixed Langevin step size of $1e-3$, and with preconditioning at a fixed step size of $0.1$.

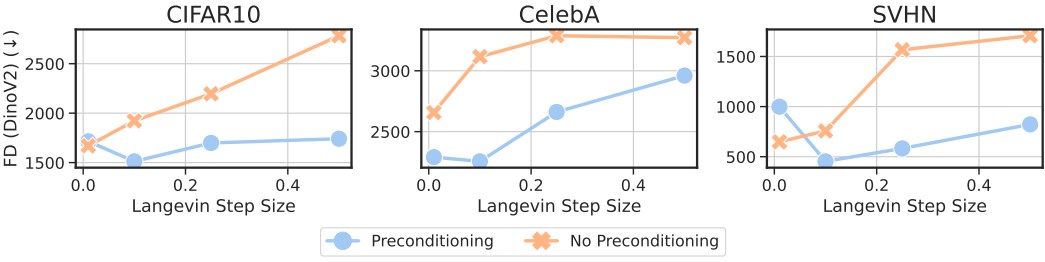

Figure 3: FDD for Langevin PC models with and without preconditioning across different step-sizes. Models trained with preconditioned Langevin dynamics experience significantly less degradation in sample quality at higher step-sizes. Corresponding FID graphs may be found in Appendix A.3

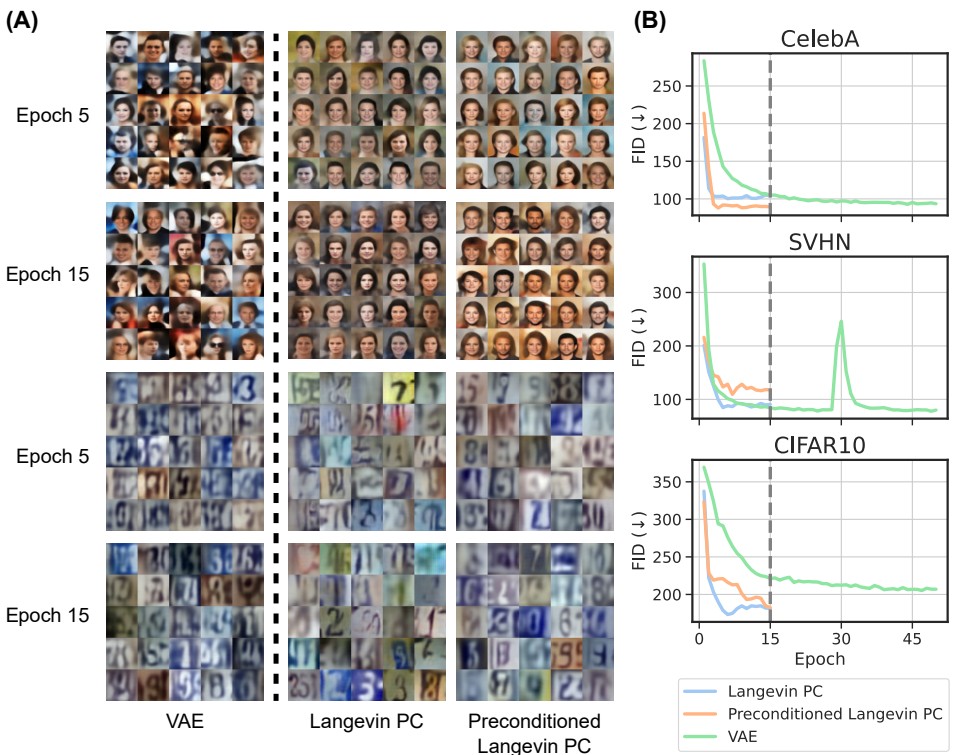

Figure 4: (A) Samples from identical generative models trained as VAEs (left), with LPC (middle), and with preconditioned LPC (right) on CelebA 64x64 (top), and SVHN (bottom). Epoch 50 samples for VAE models can be found in Appendix A.3. (B) Sample FID curves of VAE and LPC models throughout training, showing that LPC generally converges in significantly fewer epochs than their equivalent VAE trained models, with certain models converging in as few as 3 epochs.

| Model | Dataset | FID (↓) | FDD (↓) | Density (↑) | Coverage (↑) |
|---|---|---|---|---|---|
| LPC (Epoch 15) | CelebA | 104.14 | 2229.29 | 0.03 | 0.02 |
| Preconditioned LPC (Epoch 15) | | **89.29** | 2144.65 | 0.15 | 0.02 |
| VAE (Epoch 15) | | 104.97 | **1718.71** | **0.19** | **0.06** |
| VAE (Epoch 50) | | 93.53 | 1784.89 | 0.12 | 0.06 |
| LPC (Epoch 15) | SVHN | 90.66 | 494.87 | **0.14** | **0.33** |
| Preconditioned LPC (Epoch 15) | | 117.79 | **453.41** | 0.09 | 0.30 |
| VAE (Epoch 15) | | 84.21 | 557.27 | 0.06 | 0.14 |
| VAE (Epoch 50) | | **79.99** | 547.71 | 0.05 | 0.15 |
| LPC (Epoch 15) | CIFAR10 | **181.46** | 1552.95 | 3.25 | 0.06 |
| Preconditioning LPC (Epoch 15) | | 182.58 | **1512.27** | **4.65** | 0.06 |
| VAE (Epoch 15) | | 221.23 | 1641.05 | 3.76 | 0.05 |
| VAE (Epoch 50) | | 206.95 | 1548.85 | 3.97 | 0.06 |

Table 1: FID, FDD, Precision and Density values for Langevin PC (LPC), preconditioned LPC and VAE models. Note that precision and density values are also computed using Dino representations, as recommended in (Stein et al., 2023). VAE models were trained for 50 epochs to ensure convergence.

### 3.3 Samples and Metrics

We train identical generative models using the standard VAE objective, alongside the LPC and pre-conditioned LPC methodologies described herein. Models were furthermore trained with identical SGD hyperparameters, including learning rate, optimizer and batch size to ensure a fair and like-for-like comparison. Full experimental details may be found in Appendix A.1.

We observe competitive or better performance for LPC models comparative to their VAE counter-parts. With preconditioned LPC models out-performing VAE models trained for more than 3 times as many SGD iterations (50 epochs vs 15), on SVHN and CIFAR10, in terms of FDD and density. We note that we observe this despite adopting model architectures previously validated to work well with VAE objectives, and with no extensive finetuning of LPC hyperparameters, such as Langevin chain length and preconditioning decay rate.

## 4 Conclusion

We have presented an algorithm for training generic deep generative models that builds upon the predictive coding framework of computational neuroscience and consists of three primary components: an unadjusted overdamped Langevin sampling, an amortised warm-start model, and an optional light-weight diagonal preconditioning. We have evaluated three different objectives for training our amortised warm-start model: the forward KL, reverse KL and the Jeffrey's divergence, and found consistent improvements when using the reverse KL and Jeffrey's divergence over baselines with no warm-starts (Figure 2). We have also evaluated our proposed form of adaptive preconditioning and observed an increased robustness to increaing Langevin step size (Figure 3). Finally, we have evaluated the resultant Langevin PC algorithm by training like-for-like models with the standard VAE methodology or the proposed Langevin PC algorithm, using fixed and identical hyperparameters governing the SGD learning process for both. We have observed comparative or improved performance in a number of key metrics including sample quality and diversity (Table 1), while observing training convergence in a fraction of the number of SGD training iterations (Figure 4B).

Our work opens doors in two different directions. The first is in regards to PC as an instantiation of the Bayesian brain hypothesis and as a candidate computational theory of cortical dynamics. In this setting, the introduction of Gaussian noise into the predictive coding framework may represent more than simply an implementational detail associated with Langevin sampling but rather a deeper phenomena rooted in the ability of biological learning systems such as the brain to utilise sources of endogenous noise to their advantage. It is well known that neuronal systems, including their dynamics and responses, are rife with noise at multiple levels (Faisal et al., 2008; Shadlen & Newsome, 1998). These sources of noise arise from, amongst other things, stochastic processes occuring at the sub-cellular level, impacting neuronal response through, for example, fluctuations in membrane-potential (Derksen & Verveen, 1966). Yet the precise role of such randomness, in information processing, continues to be an open question (McDonnell & Ward, 2011; Deco et al., 2013). The Langevin PC algorithm suggests one such role may be in the principled exploration of the latent space of hypotheses under one's generative model.

Secondly, from the perspective Langevin PC in-silico as a generative modelling algorithm we note a number of interesting avenues that we have not had the time to explore here. These include:

- Models with a hierarchy of stochastic variables, such as those found in most state of the art VAE models (Child, 2021; Vahdat & Kautz, 2021; Hazami et al., 2022). Which may require adopting a corresponding top-down hierarchical warm-start model.

- Automatic convergence criteria for determining when our Markov chain has converged to a certain level of error (Roy, 2020).

- Underdamped Langevin dynamics, which incorporate auxiliary momentum variables into the Langevin sampling to achieve an accelerated rate of convergence (Cheng et al., 2018; Ma et al., 2019).

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

# A  APPENDIX

## A.1  EXPERIMENTAL DETAILS

All experiments in this paper adopted the following network architectures for the generative model and approximate inference models. These models are derived from the encoder/decoder VAE architectures of (Higgins et al., 2016) with slight modifications such as the use of the SiLU activation function adopted in more recent VAE models such as (Hazami et al., 2022; Vahdat & Kautz, 2021).

| Generative Model ($\log p(\mathbf{x}, \mathbf{z}|\boldsymbol{\theta})$) | Warm-Start/Encoder Model ($\log q(\mathbf{z}|\mathbf{x}, \phi)$) |
|---|---|
| Latent Dim = 40 | Obs Dim = (64, 64) or (32, 32) or (28, 28) |
| Linear(256) | If Input = (64,64): Conv(32, 3, 3, 1)
else: Conv(32) |
| SiLU | SiLU |
| Conv(64, 4, 1, 0) | Conv(32) |
| SiLU | SiLU |
| Conv(64) | Conv(64) |
| SiLU | SiLU |
| Conv(32) | If Obs Dim = (28, 28): Conv(64, 3)
else: Conv(64) |
| SiLU | SiLU |
| If obs dim = (64, 64): Conv(32)
else if obs dim = (28, 28): Conv(32, 3, 1, 0)
else: Conv(32, 3, 1, 1) | Conv(256, 4) |
| SiLU | SiLU |
| Conv(3) | Linear(2*40)
(Softplus(beta=0.3) applied to variance component) |

Table 2: Layer argument definitions are Conv(Number of out channels, kernel size, stride, padding), and Linear(Output dimensions) for 2d convolution and linear layers respectively. Kernel size, stride and padding are 4x4, 2, and 1 respectively if not explicitly stated.

| Hyperparameter | Value |
|---|---|
| Optimizer | Adam |
| Learning Rate ($\alpha$) | 1e-3 |
| Batch size | 64 |
| Output Likelihood | Discretised Gaussian |
| Max Sampling Steps ($T$) | 300 |
| Preconditioning Decay Rate ($\beta$) | 0.99 |

Table 3: Default hyperparameters used in experiments unless explicitly stated. Note: some of these are varied as part of ablation tests, see main text for more details.

## A.2  LOW-DIMENSIONAL PROJECTION OF INFERENCE AND SAMPLING TRAJECTORIES

The problem of visualising high-dimensional trajectories is a well-known one which generally arises in the context of visualising the stochastic gradient descent trajectories of high-dimensional weights in neural networks (Gallagher & Downs, 2003; Li et al., 2017; Lipton, 2016).

Here we adapt the method suggested by (Li et al., 2017) to visualise the inference or sampling trajectories of our latent states $\mathbf{z}^{(t)}$. We apply principle component analysis (PCA) to the series of vectors pointing from our final state to our intermediate states, i.e. $[\mathbf{z}^{(1)} - \mathbf{z}^{(T)}, \ldots, \mathbf{z}^{(T-1)} - \mathbf{z}^{(T)}]$, and project our trajectories on the first two principle components. We visualise the projected trajectories on top of the loss landscape of the negative potential (log joint probability) by evaluating our generative model across a grid of latent states linearly interpolated in the direction of the principle components around the final state.

Projections of an example batch of sampling trajectories can be seen in Figure 5.

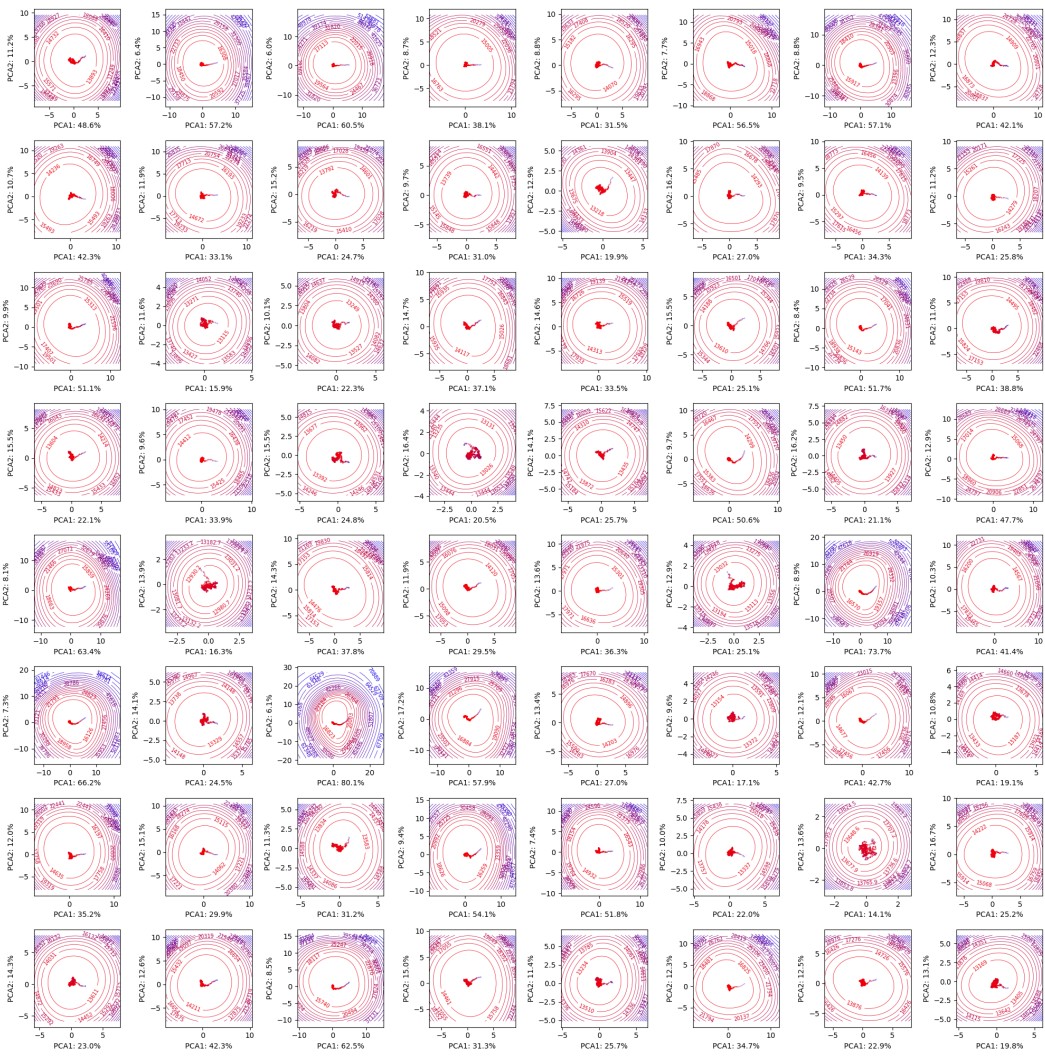

Figure 5: Projection of a 64 sample batched high-dimensional latent state trajectories under Langevin PC sampling. Contour lines and hue correspond to values of the negative log joint probability (blue high, red low), marker brightness corresponds to time-step (earlier is lighter).

## A.3 ADDITIONAL SAMPLES AND FIGURES

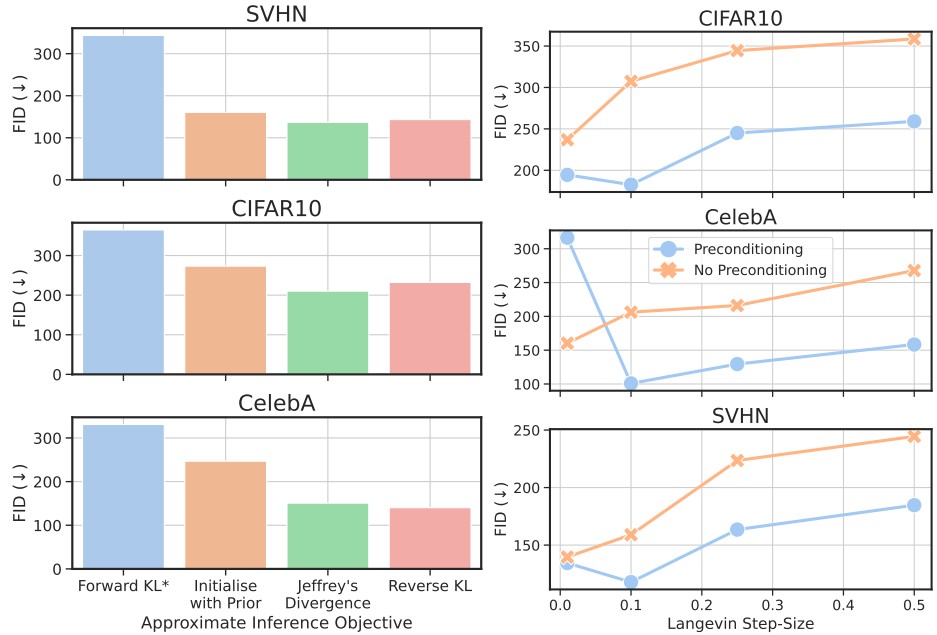

Figure 6: (Left) Fréchet inception distance, when using amortised warm-starts trained with three approximate inference objectives. Baseline is with no warm-start model, using chains initialised with the prior. (∗) Note values for forward KL objective are reported only after 1 epoch due to the instability of this objective resulting in exploding gradients. (Right) Fréchet inception distance, for Langevin PC models with and without preconditioning across different step-sizes. Models trained with preconditioned Langevin dynamics experience significantly less degradation in sample quality (FID) at higher step-sizes.

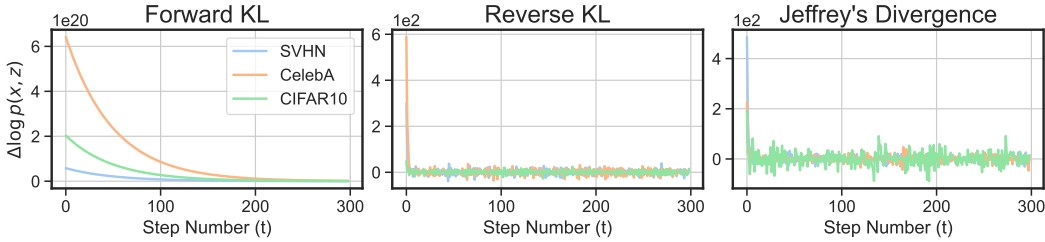

Figure 7: Log probability changes during Langevin sampling for samples from training batch 600 for our three approximate inference objectives.

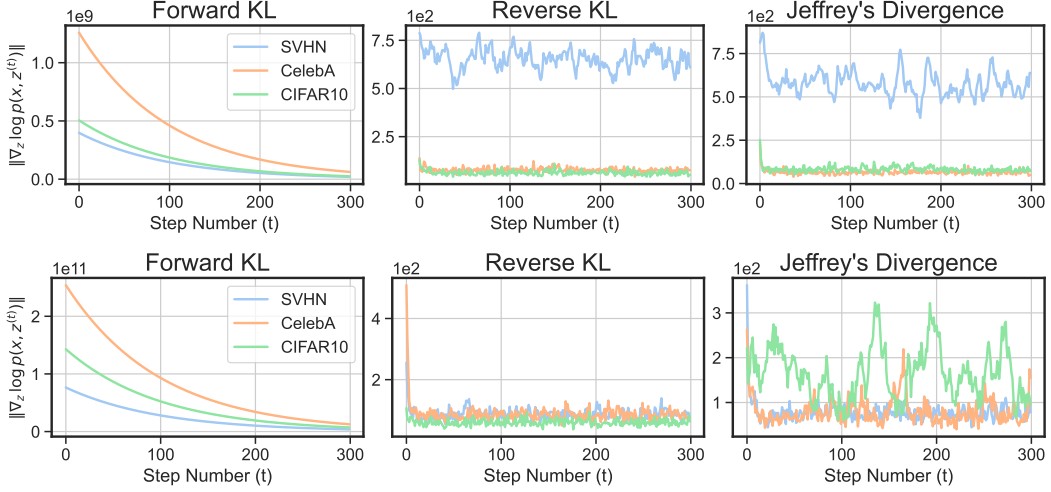

Figure 8: L2 normed log probability during Langevin sampling for samples from training batch 300 (top) and 600 (bottom), for our three approximate inference objectives.

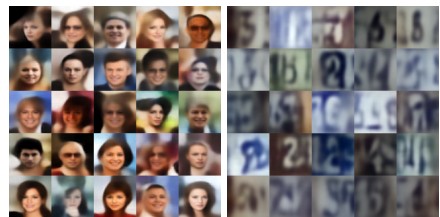

Figure 9: Epoch 50 samples from VAEs trained on CelebA 64x64 (left), and SVHN (right)

