# OpenReview forum: "Sample as you Infer: Predictive Coding with Langevin Dynamics"
_ICLR.cc/2024/Conference — ICLR 2024 Conference Withdrawn Submission_

### Official Review · Reviewer_kPGx · 2023-10-28

**Soundness:** 2 fair
**Presentation:** 4 excellent
**Contribution:** 3 good
**Rating:** 5
**Confidence:** 3

**Summary:**

This paper proposes training generative models within the predictive coding framework from computational neuroscience. Specifically, they adapt techniques including Langevin sampling with amortised inference for a “warm-start”, and preconditioning (from optimisation literature) to train latent generative models on several image datasets.

**Strengths:**

This paper provides thorough justification for its methodology with theory. It investigates and compares several different approaches training the amortised warm-start model. There are several important ablations done (such as the diagonal preconditioning).

The experiment showing improved robustness to increased langevin step size when using preconditioning, across several image datasets, is important and interesting.

**Weaknesses:**

See questions.

**Questions:**

I’m not sure I agree with the proposed set-up of comparing standard VAEs and the models proposed in this paper using identical hyperparameters for SGD. I think a more fair comparison would tune SGD hyperparameters independently for each of these models, or at least use SGD hyperparameters that were optimised for the baseline, and not necessarily for your model.

Do you have wall-clock comparisons of training time? I feel the comparison between your models taking many fewer SGD iterations to train is unfair, since each SGD iteration takes many gradient steps to sample from the models. For the same reason, I feel the comparisons done in Figure 4 are unfair. Holding the training epoch constant and comparing sample quality across models is not a fair comparison, since your LPC models compute gradients several times more per SGD iteration than the VAE models.

Why is there an asterisk in the caption of Figure 2A?

---

### Official Review · Reviewer_9aXX · 2023-10-31

**Soundness:** 2 fair
**Presentation:** 3 good
**Contribution:** 2 fair
**Rating:** 3
**Confidence:** 3

**Summary:**

The paper presents an algorithm to train generative models based on discretized langevin dynamics and predictive coding. The algorithm consists of updating thelatent variable in a generative model using langevin dynamics. To have a favorable initialization of the Langevin iterations, the authors propose to use a VAE to pick the initialization. Finally the predictive coding step in which the latent variable is updated is done using an Adam-like preconditioning.

The quality of the samples produced by the trained model is assessed against a model trained without preconditioning and against a VAE. The performances are found to be comparable.

**Strengths:**

The proposed training method goes beyond the pointwise estimation of the latent variables that is typical of predictive coding. The langevin in fact,  in principle allows to sample from the posterior of the latent variables given the data and the parameters.

**Weaknesses:**

1. The main weakness lies in the fact that the proposed algorithm is not compared to vanilla predictive coding training. Without this comparison it is impossible to establish if adding the randomness actully brings any benefit.
2. While it is said that fewer epochs are needed to learn, no assessment of the computational cost of running several steps of langevin dynamics for every parameter step is done and the need to use a VAE each time langevin iterations have to be initialized.  A comparison of the training times of the three algorithms is necessary.
3. The authors claim that to compare fairly the three algorithms (PC langevin with preconditioning, VAE, and PC langevin without preconditioning) they use the same hyperparameters. I believe that since the three algorithms are different, the hyperparameters should have been optimized separately for each algorithm. Also it is not reported how these hyperparameters were chosen.

**Questions:**

1. How is the VAE trained in your experiments? In particular, which loss is minimized?
2. Have you tried using the Metropolis adjusted Langevin dynamics, which is the proper dynamics to asymptotically sample from a probability measure?
3. How did you choose the hyperparameters you used?

---

### Official Review · Reviewer_5e67 · 2023-11-04

**Soundness:** 3 good
**Presentation:** 3 good
**Contribution:** 2 fair
**Rating:** 3
**Confidence:** 4

**Summary:**

The paper proposes a method for model learning which combines variational inference and Langevin diffusion and builds upon the predictive coding (PC) framework. An adaptive preconditioning scheme is proposed to increase the efficiency of Langevin sampling. The proposed method shows results on par with or exceeding variational autoencoders (VAEs) on image generation tasks.

**Strengths:**

- **The main ideas of the paper are presented clearly.** The paper takes ideas from predictive coding, variational inference and Langevin diffusion, and the authors present the connections between these ideas in an illuminating way. **The adaptive preconditioning method for Langevin steps also seems quite novel.**

**Weaknesses:**

- **No discussion of related works is provided/important references are missing.** Despite citing many recent developments in the fields of predictive coding and gradient-based Monte Carlo methods, no further discussion is made on how this work is connected to and/or provides novel perspectives compared to past work. As a consequence, I believe several important references are missing, for example (Hoffman 2017) is the original paper that explored the practical use of MCMC methods for deep latent variable models. In particular, (Hoffman 2017) also proposed using an inference network to initialize the chain. More recently, (Taniguchi 2022) proposed a similar model with Langevin dynamics, but where the Monte Carlo steps are done in parameter space. Their work included results on the same datasets (CelebA, SVHN, CIFAR10) and showed uniform improvement on the VAE counterparts.
- **Novelty of the proposed method is questionable.** The authors proposed several different methods for training the encoder network: forward KL, Jeffery’s divergence, backward KL. Both the discussion and experiment results show that the backward KL is most effective at learning the encoder, and this method is adopted for the remaining experiments. However, the backward KL objective is just the standard ELBO objective used in VAEs, making the proposed method an augmented version of VAE training. Since the proposed method run a Markov Chain from the initial distribution proposed by the encoder, it should produce a more accurate variational posterior at the cost of more computation. In this sense, it is not surprising that the model performs better (at least in achieving higher ELBOs). Maybe a more fair comparison would be to the importance weighted autoencoder (IWAE), since it also takes multiple posterior samples for a single input datapoint.

References:

Taniguchi, Shohei, et al. "Langevin Autoencoders for Learning Deep Latent Variable Models." *Advances in Neural Information Processing Systems* 35 (2022): 13277-13289.

Hoffman, Matthew D. "Learning deep latent Gaussian models with Markov chain Monte Carlo." *International conference on machine learning*. PMLR, 2017.

**Questions:**

- In equations (6) and (7) should it be a minus sign instead of a plus sign between the two terms?

---

### Official Review · Reviewer_LejG · 2023-11-06

**Soundness:** 3 good
**Presentation:** 3 good
**Contribution:** 2 fair
**Rating:** 5
**Confidence:** 4

**Summary:**

This work is pointed as using an adjusted version of gradient-based Langevin sampling for training a generative model. This is quite similar to SGLD (used for sampling from the posterior distribution of neural network parameters), but assumes access to a dataset to model. In order to use Langevin sampling to train a generative model, a couple of improvements are made to improve chain performance. First, the mixing time is improved via training an approximate inference model to over the dataset to warm start the chain. Different mechanisms of training the approximate inference model are explored (fwrd/rev KL + Jeffreys). A preconditioner is also formulated with respect to the Adam optimizer, reminiscent of [1] for RMSProp.

Performance of multiple divergences for training the warm-start model are compared, and reverse KL is ultimately chosen.
Further experiments are done on CelebA, CIFAR10, and SVHN to show quality of generative model bootstrapped with inference model and preconditioner vs VAE

**Strengths:**

I'm generally fine with this, SGLD has been used a lot for training Bayesian neural networks, so most of the literature is centered around how to overcome that particular problem (no dataset). This feels like a hole in the literature that is getting plugged, which is good.

* ULD for training generative models
* Approximate inference warm start network
* Interesting Adam-based preconditioning, figure 3 makes sense if preconditioner is working.
* Models converge faster on high-dim experiments

**Weaknesses:**

I think the method is fine enough. What's really bothering me is the high-dim experiments and their scores -- what are/were the limitations here?
* Comparing only against the VAE seems like its just not enough, especially when the results compared to the VAE is not especially convincing.
* Preconditioner does not always help, and when it does its not by that much. Unclear why and when it is supposed to help empirically.
All things considered, I'm not very convinced that the method is beneficial given the performance on the chosen datasets. For such numbers I would have expected more exploration or discussion of the results, and what the possible benefits could be.

**Questions:**

1. What are the computational limitations of this method?
2. What makes SVHN so difficult for this method?
3. I don't have any intuition for why FDD scores should be better/worse in this context. LPIPS is the usual score for perceptual quality.

---

### Official Review · Reviewer_x7AB · 2023-11-09

**Soundness:** 3 good
**Presentation:** 4 excellent
**Contribution:** 2 fair
**Rating:** 3
**Confidence:** 4

**Summary:**

This paper introduces a novel sampling algorithm inspired by the predictive
coding framework from neuroscience. Where with noise an overdamped Langevin
sampler is obtained which can be used to compute the gradient of the ELBO for a model.
This allows it to be used with a training regime for a deep generative model,
obtaining results competitive with VAE-based algorithms.

**Strengths:**

The paper shows that the ideas in the predictive coding framework have
quite practical applications and consequences in machine learning. The
method is explained very clearly and the experiments are well-chosen
to show that it is indeed competitive with VAEs on realistic datasets.

**Weaknesses:**

That the predictive coding framework when noise is added can be
treated as overdamped Langevin sampling is a novel and
interesting result. What is less clear is if it's significant
enough as the result seems to be algorithm that's only competitive
with well-established VAE algorithms.

While the method is clear, quite a bit of the paper is spent on
background material. Much of which is not even used in the rest
of the paper. For example, lots of space is spent on discussing
all the different divergences that could be used for optimising
the inference network. But in the end, reverse KL is used since
it works best. This is consistent with what is used for VAE
already and it feels that space could have been used on the more
novel aspects of this work.

I would have really liked a bit more motivation for why someone
who prefers to use VAEs for training deep generative models should
use this algorithm instead.

**Questions:**

Why should someone use Langevin PC over a more conventional VAE?
Could this preconditioner be applied to a VAE regime?